



# The importance of spatio-temporal snowmelt variability for
# isotopic hydrograph separation in a high-elevation catchment
Jan Schmieder[1], Florian Hanzer[1], Thomas Marke[1], Jakob Garvelmann[2], Michael Warscher[2],
Harald Kunstmann[2] and Ulrich Strasser[1]
[1]Institute of Geography, University of Innsbruck, Innsbruck, 6020, Austria
[2]Institute of Meteorology and Climate Research - Atmospheric Environmental Research, Karlsruhe Institute of
Technology, Garmisch-Partenkirchen, 82467, Germany
*Correspondence to*: Jan Schmieder (Jan.Schmieder@uibk.ac.at)
**Abstract.** Seasonal snow cover is an important temporary water storage in high-elevation regions. Especially in
remote areas, the available data is often insufficient to explicitly quantify snowmelt contributions to streamflow.
The unknown spatio-temporal variability of the snowmelt isotopic content, as well as pronounced spatial
variations of snowmelt rates lead to high uncertainties in applying the isotopic hydrograph separation method.
This study presents an approach that uses a distributed snowmelt model to support the traditional isotopic
hydrograph separation technique. The stable isotopic signatures of snowmelt water samples collected during two
spring 2014 snowmelt events at a north- and a south-facing slope were volume-weighted with snowmelt rates
derived from a distributed physics-based snow model in order to transfer the measured plot-scale isotopic
content of snowmelt water to the catchment scale. The observed $\delta^{18}O$ values and modelled snowmelt rates
showed distinct inter- and intra-event variations, as well as marked differences between north- and south-facing
slopes. Accounting for those differences, two-component isotopic hydrograph separation revealed snowmelt
contributions of 35±3 % and 75±14 % for the early and peak melt season, respectively. Differences to formerly
used weighting methods (e.g. using observed plot-scale melt rates) or considering either the north- or south-
facing slope were up to 5 and 15 %, respectively.
## 1 Introduction
The seasonal snow cover is an important temporary water storage in alpine regions. For water resources
management, the timing and amount of water released from this storage is important to know, especially in
downstream regions where the water is needed (drinking water, snow making, hydropower, irrigation water) or
where it represents a potential risk (flood, drought). In many headwater catchments, seasonal water availability is
strongly dependent on cryospheric processes and understanding these processes becomes even more relevant in a
changing climate (APCC, 2014; IPPC, 2013; Weingartner and Aschwanden, 1992). Environmental tracers are a
tool to investigate the relevant processes, but scientific studies are still rare for high-elevation regions because of
the restricted access and high risk for field measurements in these challenging conditions.
Two-component isotopic hydrograph separation (IHS) is a technique to separate streamflow into different time
source components (event water, pre-event water) (Sklash et al., 1976). The event component depicts water that
enters the catchment during an event (e.g. snowmelt) and is characterized by a distinct isotopic signature,
whereas pre-event water is stored in the catchment prior to the onset of the event and is characterized by a
different isotopic signature (Sklash and Farvolden, 1979; Sklash et al., 1976). The technique dates back to the
late 1960s (Pinder and Jones, 1969) and was initially used for separating storm hydrographs in humid



catchments. The first snowmelt-based studies were conducted in the 1970s by Dincer et al. (1970) and Martinec
et al. (1974). These studies showed a large pre-event water fraction (>50 %) of streamflow and changed the
understanding of the processes in catchment hydrology fundamentally (Klaus and McDonnell, 2013; Sklash and
Farvolden, 1979) and forced a paradigm shift, especially for humid temperate catchments. However, other
snowmelt-based studies (Huth et al., 2004; Liu et al., 2004; Williams et al., 2009) reveal a large contribution of
event water (>70 %), i.e. in permafrost or high-elevation catchments, depending on the system state (e.g. frost
layer thickness and snow depth), catchment characteristics and runoff generation mechanisms.
Klaus and McDonnell (2013) highlighted the need for accounting and quantifying the spatial variability of the
isotope signal of event water, which is still a vast uncertainty in snowmelt-based IHS. In the literature
inconclusive results prevail with respect to the variation of the snowmelt isotopic signal. Spatial variability of
snowmelt isotopic composition was statistically significant related to elevation (Beaulieu et al., 2012) in a
catchment in British Columbia, Canada with 500 m relief. Moore (1989) and Laudon et al. (2007) found no
statistical significant variation in their snowmelt $\delta^{18}O$ data, due to the low gradient and small elevation range
(approximately 30 m and 290 m) in their catchments which favours an isotopically more homogenous snow
cover. The effect of the aspect of the hillslopes on isotopic variability and IHS results in topographically
complex terrain has also been rarely investigated. Dahlke and Lyon (2013) and Dietermann and Weiler (2013)
surveyed the snowpack isotopic content and showed a notable spatial variability in their data, particularly
between north- and south-facing slopes. They conclude that the spatial variability of snowmelt could be high and
that the timing of meltwater varies with the morphology of the catchment. Dietermann and Weiler (2013) also
concluded that an elevation effect (decrease of snowpack isotopic signature with elevation), if observed, is
disturbed by fractionation due to melt/refreeze processes during the ablation period. These effects most likely
superimpose the altitudinal gradient. Aspect and slope are therefore important factors that control the isotopic
evolution of the snow cover and its melt (Cooper, 2006). In contrast, there have been various studies that have
investigated the temporal variability of the snowmelt isotopic signal, e.g. by the use of snow lysimeters (Hooper
and Shoemaker, 1986; Laudon et al., 2002; Liu et al., 2004; Maulé and Stein, 1990; Moore, 1989; Williams et
al., 2009). During the ablation season the isotopic evolution of the snowpack progresses due to percolating rain
water and fractionation caused by melting, refreezing and sublimation (Dietermann and Weiler, 2013; Lee et al.,
2010; Unnikrishna et al., 2002; Zhou et al., 2008), which leads to a homogenization of the isotopic profile of the
snowpack (Árnason et al., 1973; Dinçer et al., 1970; Stichler, 1987) and an increase in heavy isotopes of
meltwater throughout the freshet period (Laudon et al., 2007; Taylor et al., 2001; Taylor et al., 2002;
Unnikrishna et al., 2002). Therefore the characterization and the use of the evolving isotopic signal of snowmelt
water instead of single snow cores is crucial for applying IHS (Taylor et al., 2001; 2002).
There have been various approaches to cope with the variability of the input signal. If one uses more than one
$\delta^{18}O$ snowmelt value for applying the IHS method, it is important to weight the values with appropriate melt
rates, e.g. measured from the outflow of a snow lysimeter. Common weighting methods are the volume-
weighted average approach (VWA), as used by Mast et al. (1995) and the current meltwater approach (CMW),
applied by Hooper and Shoemaker (1986). Laudon et al. (2002) developed the runoff-corrected event water
approach (runCE), which accounts for both, the temporal isotopic evolution and temporary storage of meltwater
in the catchment and overcomes the shortcomings introduced by VWA and CMW. This method was furthermore
deployed in several snowmelt-based IHS (Beaulieu et al., 2012; Carey and Quinton, 2004; Laudon et al., 2004;
Laudon et al., 2007).





Tracers have successfully been used in modelling studies to provide empirical insights into runoff generation
processes and catchment functioning (Birkel and Soulsby, 2015; Birkel et al., 2011; Capell et al., 2012;
Uhlenbrook and Leibundgut, 2002), but the combined use of distributed modelling and isotopic tracers in snow-
dominated environments is rare. Ahluwalia et al. (2013) used an isotope and a modelling approach to derive
snowmelt contributions to streamflow and determined differences between the two techniques of 2 %.
Distributed modelling can provide areal melt rates that can be used for weighting the measured isotopic content
of meltwater. Pomeroy et al. (2003) describe the differences of insolation between north- and south-facing slopes
in complex terrain that lead to spatial varying melt rates of the snowpack throughout the freshet period. The use
of the areal snowmelt data from models will likely reduce the uncertainty that arises from the representativeness
of measured melt rates at the plot-scale.
The overall goal of our study was to quantify the streamflow contribution from snowmelt and hence to improve
the knowledge of hydroclimatological processes in high-elevation catchments. This study aims to test a
technique that could enhance the reliability of isotopic hydrograph separation, and thus the estimation of
snowmelt contributions to streamflow by considering the distinct spatio-temporal variability of snowmelt and its
isotopic signature in a high-elevation study region. This study has the following three objectives: 1) the
estimation of the spatio-temporal variability of snowmelt and its isotopic content, 2) the quantification of its
impact on isotopic hydrograph separation (IHS) and 3) to combine a physically-based snowmelt model with
traditional IHS. Distributed melt rates provided by a surface energy balance model were used to weight the
measured isotopic content of snowmelt in order to characterize the event water isotopic content. Traditional
weighting methods (e.g. using plot-scale observed melt rates) are compared with the newly proposed approach.
This study provides an integrated approach for streamflow components evaluation based on experimental field
work (data collection) and modelling as requested in previous studies (e.g. Seibert and McDonnell, 2002).
**2 Study area**
The 98 km² high-elevation catchment of the stream Rofenache is located in the Central Eastern Alps (Oetztal
Alps, Austria), close to the main Alpine ridge. The study area has a dry inner-alpine climate. Mean annual
precipitation is 800 mm yr⁻¹, of which 44 % falls as snow. The mean annual temperature at the gauging station in
Vent (1890 m.a.s.l., reference period: 1982-2003) is 2°C. Seasonal snow cover typically lasts from October to
the end of June at the highest regions of the valley. The basin ranges in elevation from approximately 1900
m.a.s.l. to 3770 m.a.s.l.. Average slope is 25° and average elevation is 2930 m.a.s.l. (calculated from a 50 m
digital elevation model). A thin riparian zone (<100 m width) is located in the valley floor. The predominantly
south- and north-facing slopes form the main valley, which trends roughly from west to east (cf. Fig. 1).
The bedrock consists of mainly paragneiss and mica schist and is overlain by a mantle of glacial deposit and thin
soils (< 1 m). The bedrock outcrops and unconsolidated bare rocks cover the largest part (42 %) of the catchment
(CLC, 2006). Glaciers cover approximately a third of the Rofen valley area (35 %), while pastures and
coniferous forests are located in the lowest parts of the catchment and cover less than 0.5 % (CLC, 2006).
Sparsely vegetated areas and natural grassland cover 15 and 7.5 %, respectively (CLC, 2006). Besides seasonally
frozen ground at slopes on various expositions, permafrost is likely to occur at an altitude over 2600 m.a.sl. at
the north-facing slopes (Haeberli, 1975). The annual hydrograph reveals a highly seasonal flow regime. The
mean annual discharge is 4.5 m³ s⁻¹ (reference period: 1971-2009) and is dominated by snow and glacier melt





during the ablation season, which typically lasts from May to September. The onset of the early snowmelt season
in the lower part of the basin is typically in April.
**3 Methods**
**3.1 Field sampling, measurements and laboratory analysis**
The field work was conducted during the 2014 snowmelt season between the beginning of April and the end of
June. Two short-term melt events (3 days) were investigated to illustrate the difference between early spring
season melt and peak melt. Low discharge and air temperatures with a small diurnal variation and low melt rates,
as well as a snow-covered area (SCA) of about 90 % in the basin (Fig. 7a) are the boundary conditions of the
early melt event at the end of April (cf. Fig. 2b). In contrast, the peak melt period at the end of June is
characterized by high discharge and melt rates, a flashy hydrograph, high air temperatures with remarkable
diurnal variations (Fig. 2c) and a strongly retreated snowline (SCA: 66 %; cf. Fig. 7c). Both events followed dry
antecedent conditions (no observed precipitation for at least 2 days) and no precipitation during the events itself
(Fig. 2). Discharge data are available at an hourly resolution for the gauging station in Vent and meteorological
data are obtained by 20 automatic weather stations (hourly resolution) located in and around the basin (Fig. 1).
The stream water sampling for stable isotope analysis consists of pre-freshet baseflow samples at the beginning
of March, sub-daily grab samples during the two studied events and a post-event sample in July as indicated in
Fig. 2a (grey-shaded area). Snowmelt, snowpack and surface overland flow (if observed) samples were collected
at the south- (S1, S2) and north-facing slope (N1, N2), as well as on a wind-exposed ridge shown in Fig. 1 using
a snowmelt collector. At each test site a snow pit was dug to install a $0.1 \ m^2$ polyethylene snowmelt collector at
the ground-snowpack interface. The snowmelt collector consists of a pipe that drains the percolating meltwater
into a fixed plastic bag. Tests yield a preclusion of evaporation for this sampling method. Composite daily
snowmelt water samples (bulk sample) were collected in these bags and transferred to polyethylene bottles in the
field before the onset of the diurnal melt cycle. Furthermore sub-daily grab melt samples were collected to define
the diurnal variability. The pit face was covered with white styrofoam to protect it from direct sunlight. Stream,
surface overland flow and grab snowmelt water samples were collected in 20 mL polyethylene bottles. Snow
samples from snow pit layers were filled in airtight plastic bags and melted below room temperature before
refilling them in bottles. Overall, 144 samples were taken during the study period. Snow water equivalent
(SWE), snow height (HS), snow density (SD), and various snowpack observations (wetness and hand hardness
index) were observed before the onset of the diurnal melt cycle at the study plots (Fig. 1). Mean SWE was
determined by averaging five snow tube measurements within an area of $20 \ m^2$ at each site. Daily melt rates
were calculated by subtracting succeeding SWE values. Sublimation was neglected, as it contributes only to a
small percentage (~10 %) to the seasonal water balance in high altitude catchments in the Alps (Strasser et al.,

33 2008).

All samples were treated by the guidelines as proposed by Clark and Fritz (1997) and were stored dark and cold
until analysis. The $\delta^{18}O$ and $\delta D$ content was measured with cavity ring-down spectroscopy (Picarro L1102-i).
The mean laboratory precision (replication of 8 measurements) for all measured samples was 0.06 ‰ for $\delta^{18}O$.
Due to the covariance of $\delta^2H$ ($\delta D$) and $\delta^{18}O$ (Fig. 3) all analyses were made with oxygen-18 values. Results are
expressed in the delta notation as parts per thousand relative to the Vienna Standard Mean Ocean Water
(VSMOW2).





**3.2 Model description**
For the simulation of the daily melt rates, the non-calibrated, distributed, and physically-based
hydroclimatological model AMUNDSEN (Strasser, 2008) was applied. Model features include interpolation of
meteorological fields from point measurements (Marke, 2008; Strasser, 2008); simulation of short- and
longwave radiation, including topographic and cloud effects (Corripio, 2003; Greuell et al., 1997);
parameterization of snow albedo depending on snow age and temperature (Rohrer, 1992); modelling of forest
snow and meteorological processes (Liston and Elder, 2006; Strasser et al., 2011); lateral redistribution of snow
due to gravitational (Gruber, 2007) and wind-induced (Helfricht, 2014; Warscher et al., 2013) processes; and
determination of snowmelt using an energy balance approach (Strasser, 2008). Besides having been applied for
various other Alpine sites in the past (Hanzer et al., 2014; Marke et al., 2015; Pellicciotti et al., 2005; Strasser,
2008; Strasser et al., 2008; Strasser et al., 2004), AMUNDSEN has recently been set up and extensively
validated for the Oetztal Alps region (Hanzer et al., 2016). This setup was also used to run the model in the
presented study for the period 2013–2014 using a temporal resolution of 1 hour and a spatial resolution of 50
meters. In order to determine the model performance during the study period, catchment-scale snow distribution
by satellite-derived binary snow cover maps and plot-scale observed SWE data were used for the validation.
Therefore the spatial snow distribution as simulated by AMUNDSEN was compared with a set of MODIS (500
m spatial resolution) and Landsat (30 m resolution, subsequently resampled to the 50 m model resolution) snow
maps with less than 10 % cloud coverage over the study area using the methodology described in Hanzer et al.
(2016). Model results were evaluated using the performance measures BIAS, accuracy (ACC) and critical
success index (CSI) (Zappa, 2008). ACC represents the fraction of correctly classified pixels (either snow-
covered or snow-free both in the observation and the simulation). CSI describes the number of correctly
predicted snow-covered pixels divided by the number of times where snow is predicted in the model and/or
observed, and BIAS corresponds to the number of snow-covered pixels in the simulation divided by the
respective number in the observation. ACC and CSI values range from 0 to 1 (where 1 is a perfect match), while
for BIAS values below 1 indicate underestimations of the simulated snow cover, and values above 1 indicate
overestimations. At the plot-scale, observed SWE values were compared with AMUNDSEN SWE values
represented by the underlying pixel at the location of the snow course. Catchment-scale melt rates are calculated
by subtracting two consecutive daily SWE grids, not considering sublimation to be comparable to the plot-scale
observed melt rates. Subsequently, the DEM was used to calculate an aspect grid and further to divide the
catchment into two parts: grid cells with aspects ranging from $\geq 270\,°$ to $\leq 90\,°$ were classified as 'north-facing',
while the remaining cells were attributed to the class 'south-facing'. Finally these two grids were combined to
derive melt rates for the south-facing ($melt_s$) and for the north-facing slope ($melt_n$).
**3.3 Isotopic hydrograph separation, weighting approaches and uncertainty analysis**
IHS is a steady-state tracer mass balance approach and several assumptions underlie this simple principle, which
are described and reviewed in Buttle (1994) and Klaus and McDonnell (2013). The focus of this study relies on
one of those assumptions: the spatio-temporal variability of event water isotopic signature is absent or can be
accounted for. The fraction of event water ($f_e$) contributing to streamflow was calculated from Eq. (1).
$$f_e = \frac{(c_p - c_s)}{(c_p - c_e)} \tag{1}$$





The tracer concentration of the pre-event component ($C_p$) is the $\delta^{18}$O content of baseflow prior to the onset of the
freshet period, constituted mainly by groundwater and eventually by soil water which was assumed to have the
same isotopic signal. Tracer concentration $C_s$ is the isotopic content of stream water samples for each sampling
time. The isotopic compositions of snowmelt samples were weighted differently to compose the event water
tracer concentration ($C_e$). Therefore the following five weighting approaches were deployed in the analyses:
(1)  volume-weighted with observed plot-scale melt rates (VWO)
(2)  equally weighted, assuming an equal melt rate on north- and south-facing slopes (VWE)
(3)  no weighting, only south-facing slopes considered (SOUTH)
(4)  no weighting, only north-facing slopes considered (NORTH)
(5)   volume-weighted with simulated catchment-scale melt rates (VWS)
Equation (2) is the VWS approach with simulated melt rates for north- and south-facing slopes as described in
Section 3.2, where $M$ is the simulated melt rate (in mm), $\delta^{18}O$ is the isotopic content of sampled snowmelt and
subscripts $s$ and $n$ indicate north and south, respectively. For depicting $C_e$ a daily timestep ($t$) is used,
considering daily melt rates and daily bulk snowmelt isotopic content.
$$C_e(t) = \frac{M_s(t)\delta^{18}O_s(t) + M_n(t)\delta^{18}O_n(t)}{M_s(t) + M_n(t)} \qquad (2)$$
An uncertainty analysis (Eq. (3)) was performed according to the Gaussian standard error method proposed by
Genereux (1998):
$$W_{f_e} = \left\{ \left[ \frac{c_p - c_s}{(c_p - c_e)^2} W_{C_e} \right]^2 + \left[ \frac{c_s - c_e}{(c_p - c_e)^2} W_{C_p} \right]^2 + \left[ \frac{-1}{(c_p - c_e)^2} W_{C_s} \right]^2 \right\}^{1/2}, \qquad (3)$$
where $W$ is the uncertainty, $C$ is the isotopic content, $f$ is the fraction and the subscripts $p$, $s$ and $e$ refer to the
pre-event, stream and event component. The assumption of negligible errors in the discharge measurement and
the melt rates (modelled and observed) underlay this method. The uncertainty of streamflow ($W_{C_s}$) is assumed to
be equal to the laboratory precision (0.06 ‰). For the uncertainty of the event component ($W_{C_e}$), the diurnal
temporal variation of the snowmelt isotopic signal was used (0.5 ‰) and an error of 0.04 ‰ was assumed for the
pre-event component ($W_{C_p}$), which reflects the standard deviation of the two baseflow samples. IHS results
correspond to the 95 % confidence level. Spatial variations were not considered in this error calculation method
as they represent the hydrologic signal of interest.
**4 Results**
**4.1. Spatio-temporal variability of stable isotopic signature of sampled of water sources**
The quality control was performed by the $\delta^2$H-$\delta^{18}$O plot (Fig. 3) which indicates that no shift of the linear
regression line due to secondary fractionation effects (evaporation) during storage and transport of the samples
occurred. The slope of the linear regression (slope=8.5, n=144, R²=0.93) of the measurement data slightly
deviates from that of the global meteoric (slope=8) and local meteoric water line (slope=8.1) delineated by
monthly data from the ANIP (Austrian Network of Isotopes in Precipitation) sampling site in Obergurgl, which
is located in an adjacent valley (reference period: 1991-2014). The small deviation (visible in Fig. 3) of the
sampled water sources (i.e. snowpack and snowmelt) could indicate fractionation effects induced by phase





transition (i.e. melt/refreeze and sublimation). The significant differences between isotopic signatures of pre-
event streamflow and snowmelt water enabled the IHS.
Overall, the $\delta^{18}O$ values ranged from -21.5 to -15 ‰, while snowpack samples are characterized by the most
negative and pre-event baseflow samples by the least negative values. Snowpack samples show a wide isotopic
range, while streamflow samples reveal the narrowest spread, reflect a composite isotopic signal and indicate
mixing processes of the water components. Figure 4 shows the $\delta^{18}O$ data of the water samples grouped into
different categories and split into early and peak melt data. It shows the different $\delta^{18}O$ ranges and medians of the
sampled water sources (Fig. 4a), as well as marked spatio-temporal variations of the isotopic signal (Fig. 4b and
c). It is apparent that the snowpack $\delta^{18}O$ values have a larger variation compared to the snowmelt data due to
homogenization effects (Fig. 4a), as was also shown by Árnason et al. (1973), Dincer et al. (1970) and Stichler
(1987). In contrast, the median of the $\delta^{18}O$ content of snowmelt was higher than that of the snowpack, implicit in
the fractionation processes. The median of surface overland flow $\delta^{18}O$ was higher than that of snowmelt (Fig. 4a)
for the early and peak melt period. Overall, the $\delta^{18}O$ peak melt values (Fig. 4b) reveal less variation and a higher
median than the early melt values, because fractionation effects (due to melt/refreeze and sublimation) most
likely altered the isotopic content over time (cf. Taylor et al., 2001, 2002). One major finding was that the north-
facing slope $\delta^{18}O$ data reveals a larger range and a lower median compared to the opposing slope (Fig. 4c).
Samples from the wind drift influenced site (also south-exposed) were more depleted in heavy isotopes
compared to the south-facing slope samples (Fig. 4c).
In general, the average snowmelt and snowpack isotopic content was more depleted for the early melt period
(Table 1) and changed over time because fractionation was likely to alter the snowpack and its melt. It is obvious
that the isotopic evolution (gradually enrichment) on the south-facing slope took place earlier in the annual
melting cycle of snow, following a less marked isotopic change between early and peak melt and indicates a
premature snowpack concerning the enrichment of isotopes and early ripening compared to the north-facing
slope.
Table 1 shows that meltwater sampling throughout the entire snowmelt period is required to account for the
temporal variation (cf. Taylor et al., 2001, 2002). In detail, the snowpack and snowmelt $\delta^{18}O$ data highlighted a
marked spatial inhomogeneity between north- and south-facing slopes throughout the study period. The
snowpack isotopic composition from both sampled slopes was statistically different for the early melt, but not
for the peak melt (with Kruskal-Wallis test at 0.05 significance level), whereas the snowmelt $\delta^{18}O$ showed a
significant difference throughout the complete study period (Fig. 5).
Stream water isotopic content was more enriched in heavy isotopes during the early melt period and successively
became more depleted throughout the freshet period resulting in more negative values during peak melt (Table
2). The standard deviation and range of stream water $\delta^{18}O$ during early melt was higher and could be related to a
more increasing snowmelt contribution throughout the event and larger diurnal amplitudes of snowmelt
contribution compared to peak melt (Table 2, Fig. 11).
**4.2 Snow model validation and snowmelt variability**
Figure 6 shows the values for the selected performance measures based on the available MODIS and Landsat
scenes during the period March–July 2014, while Fig. 7 shows the observed and simulated spatial snow
distribution around the time of the two events. The results indicate a reasonable model performance with a



tendency to slightly overestimate the snow cover during the peak melt season (BIAS >1). In general the CSI
does not drop below 0.7 and 80 % of the pixels are correctly classified (ACC) throughout the study period.
Table 3 holds the observed and simulated SWE values at the plot-scale. The model slightly underestimates SWE
during peak melt, but generally appears to be in quite good agreement, suggesting well simulated snowpack
processes. Throughout the study period the model deviates by 13 % from the observed SWE values, but the
representativeness (small-scale effects) of SWE values represented by the respective 50 m pixel should be
considered.
Snowmelt (observed and simulated inter-daily losses of SWE) showed a distinct spatial variation between the
north-facing and the south-facing slope for the early melt (23/24 April), but less marked variations for the peak
melt (07/08 June) (Fig. 8). Relative day-to-day differences are more pronounced for the early melt season. Both
simulated and observed melt rates are higher for the peak melt event on the south-facing slope, but not for the
north-facing slope. Simulated melt intensity on the south-facing slope at the end of April was twice the rate on
the north-facing slope, while melt rates were approximately the same for the opposing slopes during peak melt.
Small-scale snowmelt variability during early melt (north-facing slope) and partly during peak melt (south-
facing slope on 07 June) probably due to micro-topographic effects caused contrasting results between simulated
and observed melt rates (Fig. 8).
**4.3 Weighting techniques and isotopic hydrograph separation**
Differences between the applied weighting techniques, induced by the high spatial variability of snowmelt
(Section 4.2), led to different event water isotopic compositions ($C_e$) used in the IHS analyses. Table 4 lists the
event water isotopic content ($C_e$) for the five deployed weighting approaches (cf. Section 3.3). The event water
component is depleted in $\delta^{18}O$ by roughly 0.3 ‰ for the second day (24 April) of the early melt event compared
to the preceding day, but inter-daily variation during the peak melt is almost absent. Especially during early melt
(23/04 to 24/04) strong deviations between observed plot-scale melt rates and distributed (areal) melt rates
obtained by AMUNDSEN occurred (Fig. 9), and led to more differing event water isotopic compositions
between the VWS and the VWO approach (Table 4).
IHS provides estimated contributions of event and pre-event water. The event water component is labelled as the
weighted snowmelt end-member. The hydrograph and the results of the IHS applied with the VWS method for
the early and peak melt event are presented in Fig. 10. Lower flow rates and higher pre-event fractions during
early melt (Fig. 10c) and vice versa for the peak melt period (Fig. 10d) are identifiable. The total runoff volume
during the peak melt period was approximately six times higher than in the early melt period. The fraction of
snowmelt (volume) estimated with the VWS approach was 35 and 75 % with calculated uncertainties (95 %
significance level) of 3 and 14 % for the early and peak melt event, respectively. Throughout the early melt
event, the snowmelt fraction increased from 25 to 44 % (Fig. 10c; Table 5). This trend mirrors the stream
isotopic content, which is descending (Fig. 10a). Event water contribution during peak melt was generally higher
but revealed a lower range (70 to 78 %; Fig. 10d). Diurnal isotopic variations of stream water are weak for both
events (Fig. 10a and b), and could not clearly obtained due to missing data at the falling limbs of the
hydrographs.
The uncertainty calculated from Eq. (3) of the IHS applied with the VWS method in the present study was higher
(14 %) for the peak melt event because the difference between isotopic content of pre-event water and event





water was less than for the first event (3 %) (cf. Table 2 and 4). This difference controls the uncertainty the most
(cf. Section 3.3).
The use of five different weighting approaches led to strongly varying estimated snowmelt fractions of
streamflow (Fig. 11). Especially the differences between the SOUTH and the NORTH approach during both
investigated events (up to 24 %), and the differences between the VWS and the VWO approach (5 %) during
early melt (Fig. 11a) are notable. Event water contributions estimated by the different weighting methods (cf.
Section 3.3) range from 21-28 % at the beginning of the early melt event up to 31-55 % at the end of the event
(cf. Fig. 11a, Table 6). Minimum event water contributions during peak melt were estimated with 60-84 % and
maxima ranged between 67-94 % for the different weighting methods (Table 6, Fig. 11b). Beside these intra-
event variations in snowmelt contribution, the volumetric variations at the event-scale were smaller and ranged
between 28 to 40 % and 66 to 90 %, for the early and peak melt event, respectively (Table 6).
Considering only spatial variations of snowmelt isotopic signatures (i.e. comparing the NORTH/SOUTH
approach with the VWE approach) for IHS lead to differences in estimated event water fractions of maximum 7
and 14 % for the early and peak melt period, respectively (Table 6). However, considering only spatial variations
of snowmelt rates (i.e. comparing the VWS/VWO approach with the VWE approach) lead to maximum
differences in event water fraction of 3 and 2 % for the early and peak melt period, respectively (Table 6).
Surface overland flow was not considered in the IHS analyses because it reflects a runoff generation process
(geographic source) and hence is not a time source component of streamflow. However, if applied, it would most
likely increase the calculated snowmelt fraction slightly. Furthermore, snowmelt samples from the wind-exposed
site were not used in the IHS analyses because this site was only sampled on the south-facing slope during early
melt and is hardly representative for the catchment due to its limited coverage. However, an incorporation of this
data would decrease the calculated snowmelt fraction by approximately 2 %.
**5 Discussion**
**5.1 Variations of streamflow**
Snowmelt is a major contributor to the hydrograph during the spring freshet period in alpine regions and
remarkable amounts of snowmelt water infiltrate into the soil and recharge groundwater (Penna et al., 2014).
During the whole study period, two major snowmelt pulses (Mid-May and beginning of June) followed four less
pronounced ones during mid-March to early May (Fig. 2a). The hydrological response followed the variations of
air temperature, as already observed by Braithwaite and Olesen (1989), because the available net-shortwave
energy mostly controls the magnitude of snowmelt (Hock, 2003) (Fig. 2a). Peak melt occurred at the beginning
of June with maximum daily temperatures and runoff, of 15 °C and 18 mm d$^{-1}$, respectively. The following high-
flows were affected by rain (Fig. 2a) and by glacier melt due to the strongly retreated snow line and snow-free
ablation area of the glaciers in July. Diurnal variations in discharge were strongly correlated with diurnal
variations of air temperature (Fig. 2a and b) with a time lag of 3-5 hours for the early melt event and 2-3 hours
for the peak melt event. These time lags are common in mountain catchments (Engel et al., 2015; Schuler, 2002).
During peak melt, the flashy hydrograph revealed less variation in the timing of peak discharge of 7 day data
(cf. Fig. 2c) compared to the early melt, as well reported by Lundquist and Cayan (2002). An inverse
relationship between streamflow $\delta^{18}O$ and discharge (and thus snowmelt contribution) was found for the early
melt event (Fig. 10a and c). Diurnal responses of streamflow $\delta^{18}O$ were slightly identified for both events, but





masked due to missing data during the recession of the hydrograph. These results confirm earlier findings of
Engel et al. (2015) who identified inverse relationships between streamflow $\delta^{18}O$ and discharge during several
24-hour events in an adjacent valley on the southern side of the main Alpine ridge, although their findings rely
on streamflow contributions from snow as well as glacier melt. The lower stream water isotopic content during
peak melt suggests a remarkable contribution of more depleted snowmelt to streamflow and therefore confirms
the results of the IHS (Section 5.4).
**5.2 Spatio-temporal variability of snowmelt and its isotopic signature**
The magnitude of snowmelt varies in catchments with complex topography (Carey and Quinton, 2004; Dahlke
and Lyon, 2013; Pomeroy et al., 2003). This was also demonstrated for the Rofen valley in the presented study
(Fig. 8, Table 3). The small-scale snowmelt variability was high, as plot-scale observed melt rates contradicted
distributed melt rates during early melt (Fig. 9), a period of the snowmelt season when snow cover processes are
typically very heterogeneous across the catchment. The peak melt period was characterised by less spatial and
day-to-day variation in observed melt rates (Fig. 8). The modeled daily snowmelt during this period was similar
for north- and south-facing slopes, likely because of higher melt rates but a smaller snow-covered area of the
south-facing slope in contrast to the north-facing slope during peak melt (Fig. 9). The model performance was
good according to SWE values (Table 3) and to snow cover extent (Fig. 6 and 7). The spatial variations of
snowpack isotopic content are significantly evidenced for north- and south-facing slopes, as also shown by
Carey and Quinton (2004), Dahlke and Lyon (2013) and Dietermann and Weiler (2013) in their high-gradient
catchments, whereas ambiguous findings exist for the spatial variation of the snowmelt isotopic signal. It is not
clear to which extent altitude is important, as Dietermann and Weiler (2013) stated that a potential elevation
effect is likely to be disturbed by melting processes (isotopic enrichment) depending on catchment morphology
during the ablation period. An altitudinal gradient was not considered in the present study, but possible effects
on IHS are discussed in Section 5.6. Beaulieu et al. (2012) detected elevation as a predictor, which explained
most of the variance they observed in snowmelt $\delta^{18}O$ from four distributed snow lysimeters. Moore (1989) and
Laudon et al. (2007) found no significant difference of $\delta^{18}O$ in their lysimeter outflows, likely due to the small
elevation gradient of their catchments which favours an isotopically homogenous snowpack, whereas
Unnikrishna et al. (2002) found a remarkable small-scale spatial variability. The difference of snowmelt (not
snowpack) isotopic signature between north- and south-facing slopes was clearly shown in the presented study.
The dataset is small, but reveals clear differences enforced by varying magnitudes and timing of melt processes
through solar radiation on the opposing slopes (cf. Fig. 5). Temporal snowmelt isotopic variability is greater for
the north-facing slope compared to the south-facing slope (Fig. 5), which was also pointed out by Carey and
Quinton (2004) in their subarctic catchment. Earlier homogenization of the snowpack isotopic profile and earlier
melt out are responsible for this phenomenon (cf. Dincer et al., 1970; Unnikrishna et al., 2002). Fractionation
processes controlled the ongoing homogenization of the snowpack between the two investigated melt events.
The isotopic homogenization of the snowpack on the south-facing slope started earlier in the melting period and
caused a smaller spatial and temporal variation compared the north-facing snowpack, as also reported by
Unnikrishna et al. (2002) and Dincer et al. (1970). However the differences between these investigated
snowpacks in the early melt season were larger than in the peak melt season. This affects IHS results, especially
because the snowmelt contributions from the south- and north-facing slope - with marked isotopic differences -
were distinct. Due to melt, fractionation processes proceeded and the snowpack became more homogenous





throughout the snowmelt season. However, inter-daily variations of snowpack isotopic content, especially for the
north-facing slope, were still observable during the peak melt period. The gradual isotopic enrichment of the
snowpack was also observable for snowmelt, as described by many others (Feng et al., 2002; Shanley et al.,
2002; Taylor et al., 2001; Taylor et al., 2002; Unnikrishna et al., 2002). Unnikrishna et al. (2002) described
significant temporal variations of snowmelt $\delta^{18}O$ during large snowmelt events (peak melt). However, these
findings could not be confirmed within in this study, probably due to the temporally limited data and should be
tested with a larger dataset.
**5.3 Validity of isotopic hydrograph separation**
The validity of IHS relies on several assumptions (Buttle, 1994; Klaus and McDonnell, 2013). The assumption –
the isotopic content of event and pre-event water differ significantly – was successfully proven, because
measured snowmelt isotopic values were markedly lower than pre-event baseflow values (cf. Table 2 and 4, Fig.
3). Spatio-temporal variations of event water isotopic content were accounted for by collecting daily and sub-
daily samples during both events throughout the freshet period and meltwater sampling at a north- and south-
facing slope, respectively. The spatially variable input of event water was considered by dividing the catchment
into two parts – a north- and a south-facing slope. This study supports the findings of Dahlke and Lyon (2013)
and Carey and Quinton (2004), emphasizing the highly variable snowpack/snowmelt isotopic content due to
enrichment in complex topography catchments. The temporal variability of event water isotopic content was
considered by bulk daily samples, which integrate the entire diurnal melting cycle. The spatio-temporal
variability of the isotopic content of pre-event water is a major limitation and could not be clearly identified due
to a lack of data and was therefore assumed to be constant. Small differences between pre-event and post-event
streamwater isotopic content support this assumption (Table 2). The assumption of soil water having the same
isotopic content as groundwater in time and space is quite critical. Some studies reveal no significant differences
(e.g. Laudon et al., 2007), whereas others do (e.g Sklash and Farvolden 1979). Isotopic differences between
groundwater and soil water were not notable due to a lack of data. Furthermore it is not known to which amount
the vadose zone contributes to baseflow in the study area. Winter baseflow used in the analyses is assumed to
integrate mainly groundwater and partly soil water. Soil water could be hypothesized to have a negligible
contribution to baseflow during winter due to the recession of the soil storage in autumn and frozen soils in
winter. The assumption – no or minimal surface storage occurs – is plausible because water bodies like lakes or
wetlands do not exist in the study catchment and due to the steep topography detention storage may not be
relevant. The transit time of snowmelt was assumed to be less than 24 h. This short travel time is characteristic
for headwater catchments with (Lundquist et al., 2005): high in-channel flow velocities; steep hillslopes; a high
drainage density with snow-fed tributaries; thin soils; most snowmelt originating from the edge of the snow-line
(small average travel distances); partly frozen soil; and observed surface overland flow. The state-of-the-art
method (runCE) to include residence times of snowmelt in the event water reservoir proposed by Laudon et al.
(2002), was applied in several IHS studies (Beaulieu et al., 2012; Carey and Quinton, 2004; Petrone et al., 2007),
but was not feasible due to the short-term character and temporally limited data of the experimental design.
**5.4 Hydrograph separation results and inferred runoff generation processes**
High contributions from snowmelt to streamflow are common in high-elevation catchments. Daily contributions
between 35 and 75 % in the Rofen valley are comparable to the results of studies conducted in other





mountainous regions, mostly outside the European Alps. Beaulieu et al. (2012) estimated snowmelt contributions
ranging from 7 to 66 % at the seasonal scale for their 2.4 km² catchment and reported contributions of 34 and
62 %, for the early melt and peak melt, respectively. The hydrograph is dominated by pre-event water during
early melt in April (Fig. 10c), which is in accordance with the results obtained by other IHS studies (Beaulieu et
al., 2012; Laudon et al., 2004; Laudon et al., 2007; Moore, 1989). Initial snowmelt events flush the pre-event
water reservoir as snowmelt infiltrates into the soil and causes the pre-event water to exfiltrate and contribute to
the streamflow. As the soil and groundwater reservoir becomes gradually filled with new water (snowmelt), the
event water fraction in the stream increases. The system is also wetter during peak melt. The dominance of event
water in the hydrograph is interpreted as an outflow of pre-event water stored in the subsurface and the gradual
replenishment of event water. The higher water table – compared to the early melt period – could cause a
transmissivity feedback mechanism (Bishop, 1991). This is a common mechanism in catchments with glacial till
(Bishop et al., 2011) and characterises higher transmissivities and hence increasing lateral flow velocities
towards to the surface. Runoff generation is spatially very variable in the study area. There are areas (meadow
patches between rock fields) were saturation excess overland flow is dominant (observed mainly at plots S1, S2
and Wind) and areas (with larger rocks and debris) were rapid shallow subsurface flow can be assumed (plot
N2). Catchment morphology controls various hydrologic processes and hence the shape of the hydrograph.
Upslope residence times of snowmelt are usually smaller due to thinner soils (observed during the field work),
steeper slopes (Sueker et al., 2000) and higher contributing areas of glaciers with impermeable ice (Behrens,
1978) and would be indicators for the more flashy hydrograph during the peak melt season. The snowmelt
contribution increased as the freshet period progressed and peaked with high contributions at the beginning of
June. Beaulieu et al. (2012) and Sueker et al. (2000) reported comparable results for their physically similar
catchments during peak melt with 62 and up to 76 % event water contributions to streamflow, respectively. At
the event-scale comparable studies are rare. Engel et al. (2015) report a maximum daily snowmelt contribution
estimated with a three-component hydrograph separation of 33 % for an 11 km² catchment southwest of the
Rofen valley with similar physiographic characteristics, but on the southern side of the main Alpine ridge. It
should be mentioned that in their study, runoff was fed by three components (snowmelt, glacier melt and
groundwater) and lower snowmelt contributions were prevalent because most of the catchment area (69 %) was
snow-free.
**5.5 Impact of spatial varying snowmelt and its δ$^{18}$O content on IHS (Assessment of weighting approaches)**
Klaus and McDonnell (2013) stress in their review paper the need for investigating the effects of the spatially
varying snowmelt and its isotopic content on IHS. The present study quantified the impact of spatially varying
snowmelt isotopic content between north- and south facing slopes on IHS results for the first time. The
difference in volumetric snowmelt contribution to streamflow at the event-scale determined using the five
different weighting methods for IHS is maximal 24 % (NORTH approach vs. SOUTH approach). The data show
that the variations between the weighting approaches (VWS, VWO and VWE) are higher throughout the early
melt season (Table 6), because small-scale variability of snowmelt and its isotopic content are more pronounced
in the early melt season. Thus the influence of spatial variability of snowmelt and its isotopic content on the
event water fraction calculated with IHS is larger during this time. Melt rates strongly differ between the south-
and the north-facing slope (Fig. 9), which was deceptively gathered by manually measured SWE, likely due to
micro-topographic effects. As the contributions from both slopes are used in Eq. (3), they strongly influence the





applied weighting technique. The weighting method SOUTH (or NORTH) represents the most extreme scenario
in which only one sampling site was deployed in the IHS analysis. Because snowmelt is more depleted in $\delta^{18}$O
and closer to pre-event water isotopic content on the south-facing slope during peak melt, this scenario has the
greatest effect on IHS and leads to the strongest deviation in estimated snowmelt fractions (up to 15 %
overestimation compared to the VWS approach). Similar to the VWE method, snowmelt isotopic data was not
volume-weighted in other studies (e.g. Engel et al., 2015) since snowmelt data was not available. This has a
more distinct effect on IHS during the early melt season because of the higher spatio-temporal variability in
snowmelt and its isotopic content compared to the peak melt season and led to a deviation in the snowmelt
fraction of 2 % and 3 % compared to the VWS and VWO approach, respectively. Although the differences seem
to be small, it should be mentioned that differing snowmelt and isotopic values offset each other in this particular
case, which led to the relatively small differences in estimated snowmelt fractions (Table 6). Nevertheless the
results of VWS are more correct for the right reason, because single observed plot-scale melt rates do not
represent distributed snowmelt contribution at the catchment-scale. Therefore one can hypothesize that
distributed simulated melt rates enhance the reliability and feasibility of IHS, whereas plot-scale weighting
implements a very high error caused by the difficulty in finding locations that represent the melt rate of a slope
in complex terrain. The IHS results of this study are more sensitive to the spatial variability of snowmelt $\delta^{18}$O,
than spatial variations of snowmelt rates (Table 6). This is even more pronounced for the peak melt period,
because snowmelt rates were similar for the north- and south-facing slope, probably due to an isothermal snow
cover throughout the catchment.

**5.6 Limitations of the study**

Collecting water samples in high-elevation terrain is challenging due to limited access and high exposure to risk
(e.g. avalanches), limiting especially high-frequency sampling. Hence some limitations are inherent in the
presented study. Potential elevation effects on snowmelt isotopic content were not tested. The opposing sampling
sites (S1-N1 and S2-N2) were at the same elevation (Fig. 1). It was assumed that the differences of north- and
south-facing slopes were significantly greater than a possible altitudinal gradient of snowmelt isotopic content.
This hypothesis was not tested, but assumed to be valid based on the results of other studies (Dietermann and
Weiler, 2013). However, accounting for a potential altitudinal gradient (decrease of snowmelt $\delta^{18}$O with
elevation) would lead to more depleted isotopic signatures of event water and hence to lower event water
fractions. A disadvantage is that no snow survey was conducted prior to the onset of snowmelt (peak
accumulation) to estimate spatial variability in bulk snow $\delta^{18}$O. Because snowmelt is used for applying IHS, it is
not clear to which degree the spatial variability of the snowpack isotopic content is important. Two-component
isotopic hydrograph separation was successfully applied using the end-members snowmelt and baseflow, but
potential contributions of glacier melt were neglected. Because glaciers in the catchment were still covered by
snow during the peak melt season, a significant contribution from ice melt was therefore assumed to be unlikely.
Nevertheless negligible amounts of basal meltwater could originate from temperate glaciers. No samples could
be collected during the recession of the hydrograph (at night). Despite spatial variability of the event water signal
was the focus of the study, only temporal variability was considered in the Genereux-based uncertainty.
Furthermore, model results and observed discharges were assumed to be free of error in the analyses. As pointed
out, instrumentation and accessibility are major problems for high-elevation studies and their sampling
strategies. For this study it turned out that composite snowmelt samples were easier to collect, representing the





day-integrated melt signal. A denser network of melt collectors would be desirable, as well as a snow lysimeter
to gain high-frequency data automatically. Representative samples of the elevation zones and different
vegetation belts could be important too, especially in partly forested catchments with a distinct relief (cf.
Unnikrishna et al., 2002).
**6 Conclusions**
The presented study provides new insights into the variability of snowmelt isotopic content and highlights its
impact on IHS in a high-elevation environment. The spatial variability of snowmelt isotopic signatures was
extensively considered by experimental investigations on south- and north-facing slopes to define tracer
concentrations of the snowmelt end-member with greater accuracy. This study clearly shows that distributed
snowmelt rates simulated by a model, fed with meteorological data from local automatic weather stations, affect
the weighting of the event water isotopic signal, and hence the estimation of snowmelt fraction in the stream by
IHS. The study provides a variety of relevant findings that are important for hydrologic research in high-alpine
environments: a distinct snowmelt variability between north- and south-facing slopes was shown for this
complex terrain, especially during the early melt season; isotopic signatures of snowmelt water were
significantly different between north-facing and south-facing slopes, which resulted in a pronounced effect on
estimating snowmelt contributions to streamflow with IHS; differences in the estimated snowmelt fraction due to
the weighting methods used for IHS were quantified by up to 24 %. It became evident that it is hardly possible to
characterize the event water signature of larger slopes based on plot-scale snowmelt measurements. Applying
distributed modelling reduced the uncertainty of the spatial snowmelt variability inherent in point-scale
observations. Hence, applying the VWS method provided more reasonable results than the VWO method.
Sampling north- and south-facing slopes is of major importance in conducting snowmelt-based IHS in
mountainous catchments with complex topography in which a non-uniform input of snowmelt can be expected.
Therefore, it has to be pointed out that the selection of sampling sites has a major effect on IHS results. Sampling
at least north-facing and south-facing slopes in complex terrain and using distributed melt rates to weight the
snowmelt isotopic content of the differing exposures is therefore highly recommended for applying snowmelt-
based IHS.
*Acknowledgments.* The authors wish to thank the Institute of Atmospheric and Cryospheric Sciences of the University of
Innsbruck, the Zentralanstalt für Meteorologie and Geodynamik, the Hydrographic Service of Tyrol and the TIWAG-Tiroler
Wasserkraft AG for providing hydrological and meteorological data, the Amt der Tiroler Landesregierung for providing the
DEM, as well as many individuals who have helped to collect data in the field.

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



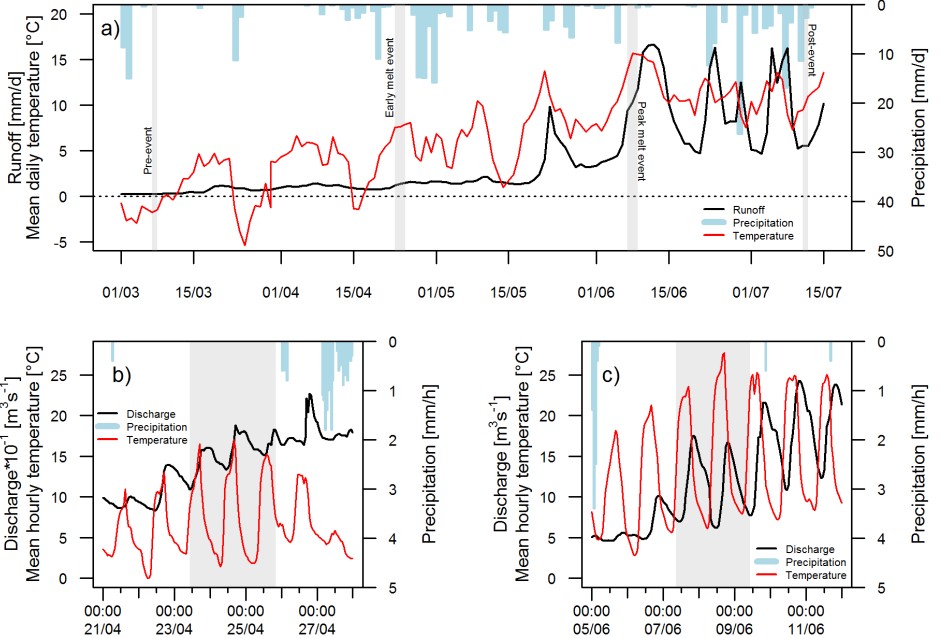

2  **Figure 2: (a) Daily precipitation, air temperature, and discharge during the complete study period; Hourly hydro-**
3  **climatologic data of a 7-day period around the (b) early melt and (c) peak melt event. Data was measured at the outlet**
4  **of the catchment. Grey-shaded areas indicate the investigated events.**




**Figure 3: Relationship between $\delta^2$H and $\delta^{18}$O values of water sources sampled during the snowmelt season 2014 in the**
**Rofen valley, Austrian Alps.**

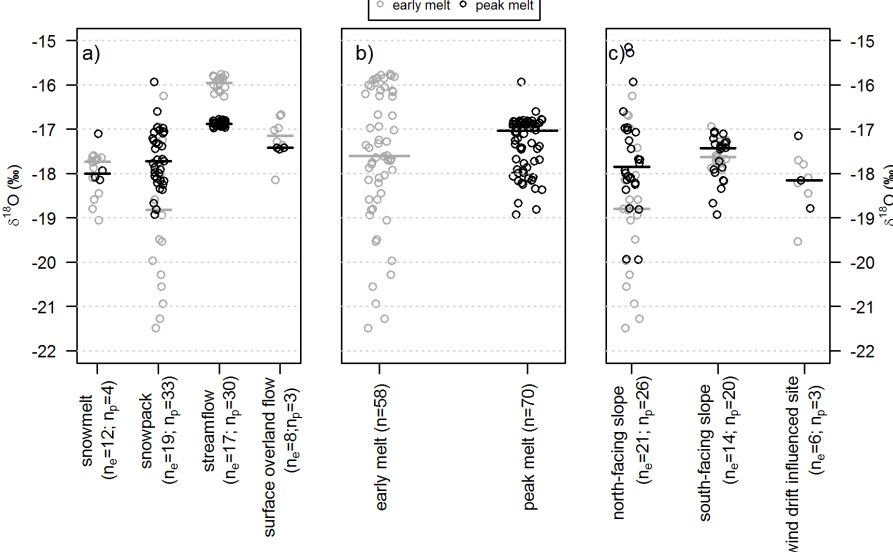

**Figure 4: 1-D scatterplots for $\delta^{18}$O of collected water samples split into (a) water sources, (b) stage of snowmelt and (c)**
**spatial origin. Grey circles indicate early melt samples and black circles are for peak melt samples. The grey and**
**black line represents the median of early and peak melt data, respectively. $N_e$ is the number of early melt samples and**
**$n_p$ is the number of peak melt samples.**

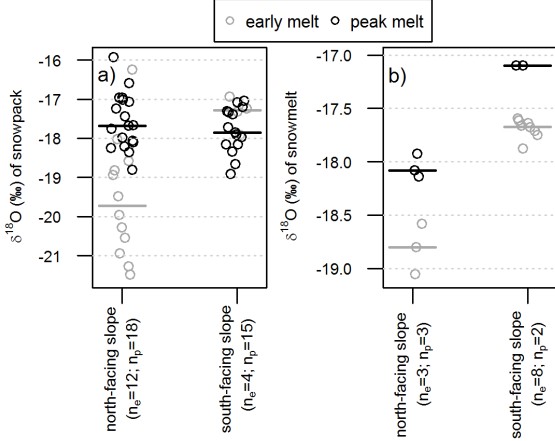

**Figure 5: 1-D scatterplots for $\delta^{18}$O of (a) snowpack and (b) snowmelt of north- and south-facing slopes. Grey circles**
**indicate early melt samples and black circles are for peak melt samples. The grey and black line indicates the median**
**of the early and peak melt data, respectively. $N_e$ is the number of early melt samples and $n_p$ is the number of peak**
**melt samples.**





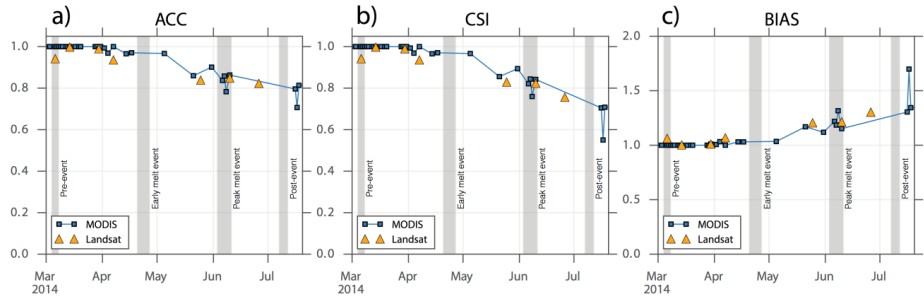

**Figure 6: Performance measures (a) Accuracy (ACC), (b) Critical Success Index (CSI), and (c) BIAS as calculated by comparing AMUNDSEN simulation results with satellite-derived (MODIS/Landsat) snow maps.**

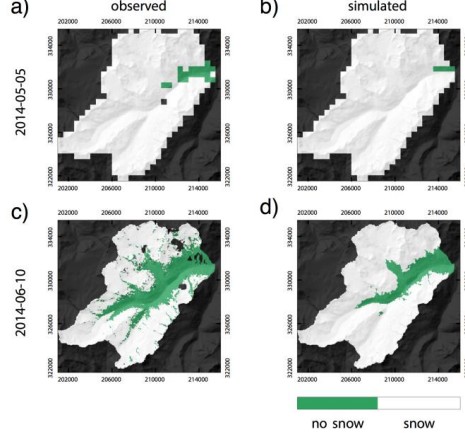

**Figure 7: Comparison of observed and simulated snow distributions for (a, b) May 5 (MODIS scene) and (c, d) June 10, 2014 (Landsat scene).**





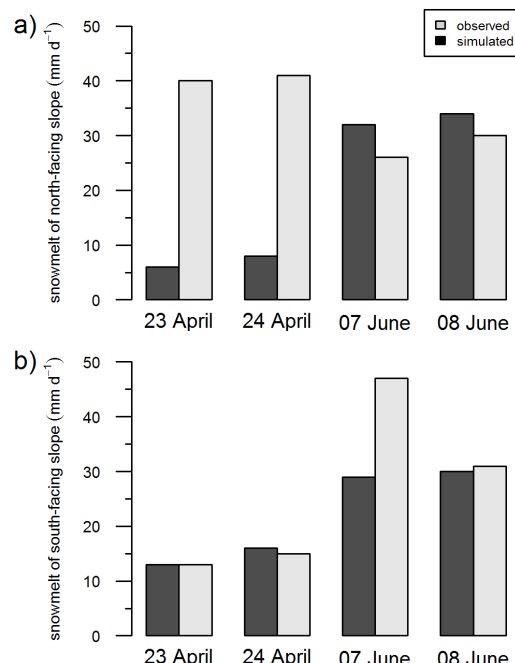

**Figure 8: Differences between the observed (plot scale) and simulated (catchment scale) daily snowmelt on (a) the**
**north-facing and (b) the south-facing slope for the early melt (23/24 April) and peak melt (07/08 June).**

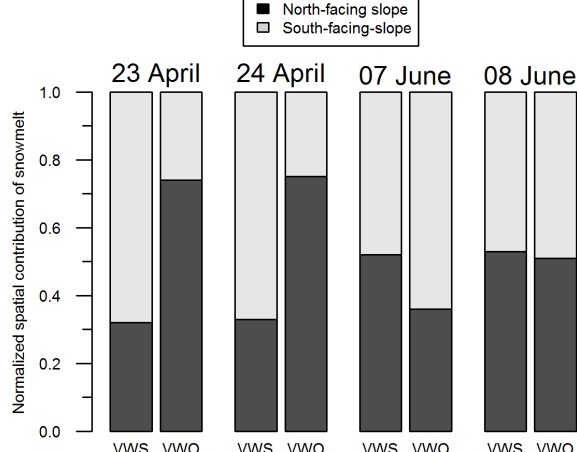

**Figure 9: Comparison of the spatial contribution of weighting approaches. VWS: volume-weighted with simulated**
**(areal) melt rates. VWO: volume-weighted with observed (plot-scale) melt rates.**




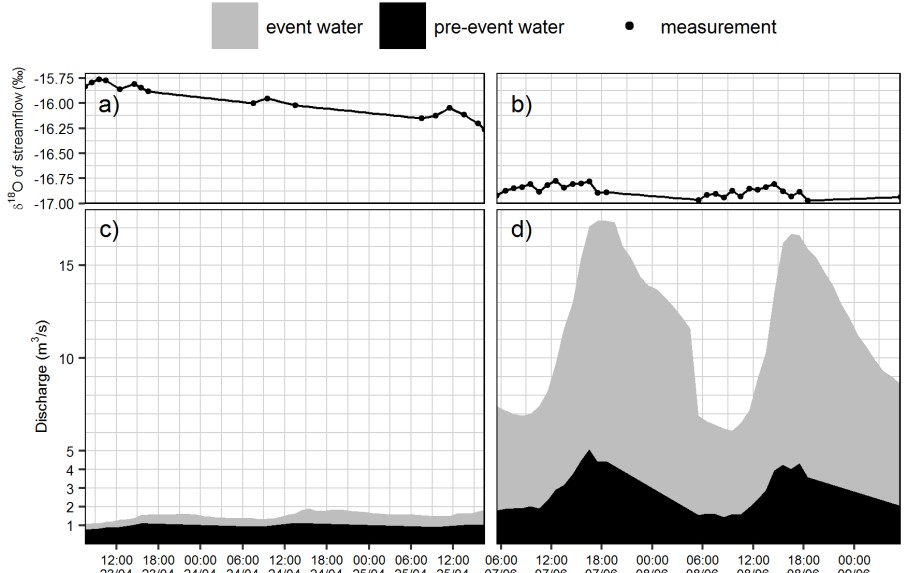

Figure 10: Linearly interpolated stream isotopic content of Rofenache for (a) the early melt and (b) the peak melt event. Dots indicate measurements. Event and pre-event water contributions during (c) the early melt and (d) the peak melt event calculated with the VWS approach.

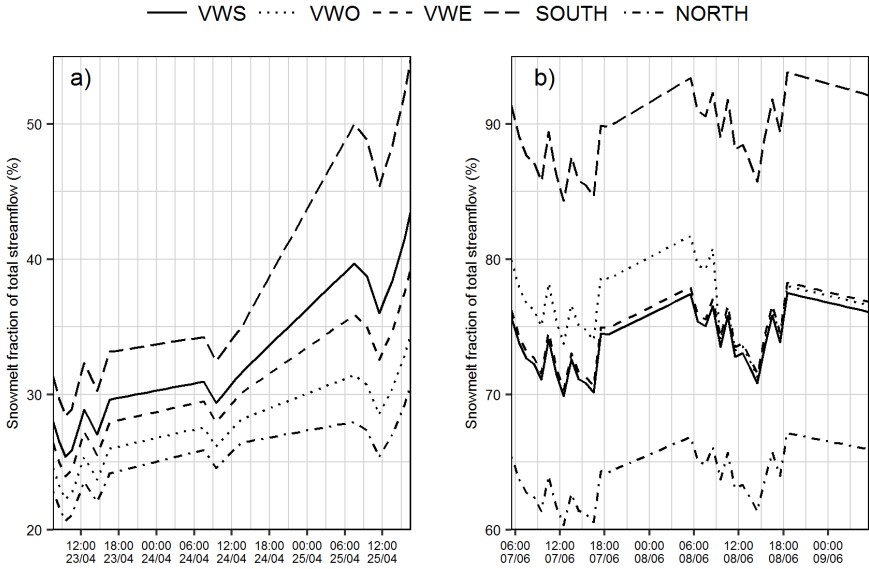

Figure 11: Comparison of weighting techniques used for estimating snowmelt fraction with IHS during (a) early melt and (b) peak melt. Scale of Y-axis in b) differs from that in a).

Table 1: Average isotopic content of snowpack and snowmelt with standard deviation for north- and south-facing slopes during the early and the peak melt event. Values are averages of three consecutive days.





|  | North-facing slope | | South-facing slope | |
|---|---|---|---|---|
|  | Snowpack $\delta^{18}$O (‰) | Snowmelt $\delta^{18}$O (‰) | Snowpack $\delta^{18}$O (‰) | Snowmelt $\delta^{18}$O (‰) |
| **Early melt event** | -19.7±0.6 (n=12) | -18.8±0.2 (n=3) | -17.3±0.3 (n=4) | -17.4±0.2 (n=8) |
| **Peak melt event** | -17.6±0.4 (n=18) | -17.9±0.1 (n=3) | -17.9±0.1 (n=15) | -17.1±0.0 (n=2) |

**Table 2: Descriptive statistics of streamflow isotopic content (Rofenache) during events of the snowmelt season 2014.**
**Data is sampled at the outlet of the basin.**

|  | **Pre-event** | **Early melt** | **Peak melt** | **Post-event** |
|---|---|---|---|---|
| Date | 07/03 | 23/04 – 25/04 | 07/06 – 09/06 | 11/07 |
| Average ($\delta^{18}$O ‰) | -15.02 | -15.97 | -16.87 | -15.09 |
| Standard deviation ($\delta^{18}$O ‰) | 0.04 | 0.16 | 0.05 | n/a |
| Range ($\delta^{18}$O ‰) | 0.05 | 0.50 | 0.20 | n/a |
| Number of samples | 2 | 17 | 30 | 1 |

**Table 3: Comparison of observed and simulated (represented by the underlying pixel) SWE values at the plot-scale.**

| Site | Date | Stage of snowmelt season | SWE [mm] | | Difference between observed and simulated SWE [%] |
|---|---|---|---|---|---|
|  |  |  | *Observed* | *Simulated* |  |
| S1 | 2014-04-23 | Early melt | 141 | 151 | 7 |
| N1 | 2014-04-23 | Early melt | 351 | 356 | 1 |
| Wind | 2014-04-24 | Early melt | 201 | 229 | 14 |
| S1 | 2014-04-25 | Early melt | 113 | 78 | -31 |
| N1 | 2014-04-25 | Early melt | 270 | 293 | 9 |
| N2 | 2014-06-07 | Peak melt | 594 | 477 | -20 |
| N2 | 2014-06-08 | Peak melt | 568 | 435 | -23 |
| N2 | 2014-06-09 | Peak melt | 537 | 390 | -27 |
| Mean deviation between observed and simulated SWE | | | | | 13 |

**Table 4: Isotopic characterization of the event water component by the applied weighting techniques**

|  | Event water isotopic composition ($\delta^{18}$O ‰) | | | |
|---|---|---|---|---|
|  | 23/04 | 24/04 | 07/06 | 08/06 |
| VWS | -17.9 | -18.2 | -17.5 | -17.5 |





| | | | | |
|---|---|---|---|---|
| VWO | -18.3 | -18.6 | -17.4 | -17.5 |
| VWE | -18.1 | -18.3 | -17.5 | -17.5 |
| NORTH | -18.6 | -18.8 | -17.9 | -17.9 |
| SOUTH | -17.6 | -17.9 | -17.1 | -17.1 |

3 **Table 5: Discharge quantities of the Rofenache for the early and peak melt event at the outlet of the basin.**

| | Event | |
|---|---|---|
| | Early Melt | Peak Melt |
| Date | 23/04 – 25/04 | 07/06 – 09/06 |
| Mean discharge | 1.5 $\mathrm{m^3\,s^{-1}}$ | 11.5 $\mathrm{m^3\,s^{-1}}$ |
| Peak discharge | 1.9 $\mathrm{m^3\,s^{-1}}$ | 17.4 $\mathrm{m^3\,s^{-1}}$ |
| Volume runoff | 3.3 mm | 20.7 mm |
| Mean event water fraction | 35±3 % | 75±14 % |
| Peak event water fraction | 44 % | 78 % |

6 **Table 6: Event water contribution to streamflow estimated with different weighting techniques. The error indicates**
7 **the variability (standard deviation) and the brackets depict the range.**

| | Event water contribution (%) | | | | |
|---|---|---|---|---|---|
| | VWS | VWO | VWE | NORTH | SOUTH |
| **Early melt event** | 35±6 (25-44) | 30±4 (22-35) | 33±5 (24-39) | 28±3 (21-31) | 40±9 (28-55) |
| **Peak melt event** | 75±2 (70-78) | 78±3 (71-82) | 76±2 (70-78) | 66±2 (60-67) | 90±3 (84-94) |

