# Peer review of "The importance of spatio-temporal snowmelt variability for isotopic hydrograph separation in a high-elevation catchment"

_Hydrology and Earth System Sciences, 2016_

## Referee Comment (RC1) · Anonymous Referee #1 · 8 Jun 2016

General comment This paper analyses the spatio-temporal variability of snowmelt and of its composition in stable isotopes in an alpine catchment and takes advantage of different weighting methods to calculate the isotopic composition of snowmelt, based on melt rates, with the aim to assess the impact of snowmelt variability on the results of two-component hydrograph separation. This is certainly an original idea that tries to overcomes the issues deriving from the highly variable isotopic composition of snowmelt both in space and in time, as largely document by literature studies. Therefore, this research is surely interesting for the readers of HESS. The manuscript is well written, logically organized, clearly structured. The Introduction leads to research questions that are developed in the text, with data that well support the results. Graphs

are very well prepared and tables are generally meaningful. I have only few major comments and indications that could be useful to address in order to increase the clarity, and thus the impact, of this work (see below).

Specific comments

2, 32-35. This is a critical part of the Introduction and should be better explained. It seems to me that what stated here is more relevant for the temporal variability rather than the spatial variability. Please, specify.

3, 17-18. The third objective seems to me more a tool than a specific objective itself. I suggest to revise it.

4, 22-24. This is interesting, and I congratulate the authors for having collected both bulk snowmelt samples and sub-daily samples to assess the diurnal variability of snowmelt. However, as far as I see, no data are presented or no discussion is reported to compare the bulk with the sub-daily samples. I encourage the authors to do so because, in my opinion, knowing which variability we miss is we sample only once at the end of the day (bulk sample) instead of taking more samples during an individual melt event at the daily scale would be of great practical interest.

4, 6. According to Figure 2, Table 3 and 4, samples were collected (and hydrograph separation was conducted) for two snowmelt events in the early melt season (23 and 24 of April) and two snowmelt events in the late melt season (7 and 8 of June) but the authors talk about 'two short-term melt events (3 days)'. This is not clear because usually diurnal-melt driven fluctuation in discharge are considered as individual melt events, so I count here 4 runoff events. More importantly, results are often presented in term of 'early melt' and 'peak melt', so, I believe, averaging or integrating the two couples of events. This operation could partly mask the intrinsic variability of the 4 events and therefore I suggest to present the data and the results separately (as, for instance, in Fig. 4).

[Figure]

4, 15-16. Two samples only to characterize baseflow and therefore the isotopic composition of pre-event water can be too little and a potential weak point for the following calculations. This aspect should be briefly discussed in the Discussion section or in section 5.6. Moreover, a recent work analyses the impact of sampling strategy of pre-event water (before the individual melt event or before the start of the freshet period) for two-component hydrograph separation during snowmelt periods. The discussion about characterization of the pre-event water isotopic composition could start from the results obtained in the recent paper by Penna et al. (2016) (see suggested additional references below). Please note that only one sample of pre-event water is visible in Fig. 3: are the two samples so similar?

6, 23-24. The diurnal temporal variation of snowmelt isotopic signal (0.5 per mil) was used for the uncertainty of the event component in the hydrograph separation, according to the traditional approach by Genereux (1998). This is fine but one aspect is not clear to me: was the same variability used for each of the 4 events? I think it is very unlikely that the 4 of them have the same diurnal temporal variability. Instead, the variability of each day should be used for the assessment of uncertainty of each runoff event. Please, fix this or explain.

6, 24. Here, the standard deviation of the two baseflow samples s relative small and it reflects, I assume, in small uncertainty values. I wonder if, having many baseflow samples, the variability would be greater and so the uncertainty of the pre-event component (see my comment above on 4, 15-16).

11, 17-18. This is, to me, contrasting to what stated at 4, 22-24. I think that while the bulk sample integrate the diurnal melting cycle it also smooths out the variability of the snowmelt signal very much. This should be tested, reported and discussed.

The figures should be introduced following an order (1, 2, 3. . .). See, for instance at 4, 7; 7, 35;

Terminology: I recommend to consistently change the term 'isotopic content' into the

[Figure]

more physically appropriate 'isotopic composition'. I also suggest to replace the term 'isotopic hydrograph separation' into 'isotope-based hydrograph separation'.

Figures 3. Use different colours or symbols for snowmelt and snowpack to better distinguish them. One pre-event samples is missing. Moreover, since there is an equation for the local meteoric water line, I suggest to plot it instead of the global meteoric water line. . .it makes more sense.

Figure 4 and Figure 5. They are quite clear but I think that the information could be conveyed much more clearly by using box-plots instead. Please, consider changing these '1-D scatterplots' (by the way, is this the right term?) into box-plots.

Figure 6 and Figure 7 are too small but this is probably due to the editorial form.

Table 1. If Figure 4 and Figure 5 are converted to box-plots this table could be probably skipped because redundant. Please, consider this possibility.

Table 2. I think that it would be more informative to report the values of the two pre-event samples individually. As I stated before, reporting the average of streamflow during the two early melt and the two late melt events is not so informative to me. Consider reporting all data in a different way (box-plots again) or even skipping this table and reporting the values of the two pre-event samples in the text.

Table 3. Could this table be incorporated as bar plot in Figure 8? Please, consider the feasibility of this suggestion.

Table 5. Why is there no uncertainty reported for the peak event water fraction? According to Genereux (1998) it can be computed. Please, fix this.

Minor comments and technical corrections

1, 11. I suggest to remove 'unknown'.

2, 39. Explain shortly which are the mentioned shortcomings.

3, 7. For consistency, 'describe' should be 'described'.

12, 15. Typo: 'were' should be 'where'.

12, 29. The title is too long, please revise.

13, 2. Is 'deployed' the right term here?

References

The correct citation for Birkel et al., 2011 is: Birkel, C., D. Tetzlaff, S. M. Dunn, and C. Soulsby (2011), Using time domain and geographic source tracers to conceptualize streamflow generation processes in lumped rainfall-runoff models, Water Resour. Res., 47, W02515, doi:10.1029/2010WR009547.

The correct citation for Capell et al., 2012 is: Capell, R., D.Tetzlaff, and C.Soulsby (2012), Can time domain and source area tracers reduce uncertainty in rainfall-runoff models in larger heterogeneous catchments?, Water Resour. Res., 48, W09544, doi:10.1029/2011WR011543.

The correct citation for Engel et al., 2015 is: Engel, M., Penna, D., Bertoldi, G., Dell'Agnese, A., Soulsby, C., and Comiti, F. (2016) Identifying run-off contributions during melt-induced run-off events in a glacierized alpine catchment. Hydrol. Process., 30: 343–364. doi: 10.1002/hyp.10577.

The correct citation for Laudon et al., 2004 is: Laudon, H., J. Seibert, S. Köhler, and K. Bishop (2004), Hydrological flow paths during snowmelt: Congruence between hydrometric measurements and oxygen 18 in meltwater, soil water, and runoff, Water Resour. Res., 40, W03102, doi:10.1029/2003WR002455.

The correct citation for Lundquist et al., 2005 is: Lundquist, J. D., M. D. Dettinger, and D. R. Cayan (2005), Snow-fed streamflow timing at different basin scales: Case study of the Tuolumne River above Hetch Hetchy, Yosemite, California, Water Resour. Res., 41, W07005, doi:10.1029/2004WR003933.

The correct citation for Strasser et al., 2004, is: Strasser, U., J. Corripio, F. Pellicciotti, P. Burlando, B. Brock, and M. Funk (2004), Spatial and temporal variability of meteorological variables at Haut Glacier d'Arolla (Switzerland) during the ablation season 2001: Measurements and simulations, J. Geophys. Res., 109, D03103, doi:10.1029/2003JD003973.

Additional relevant references that should be considered are:

Earman, S., A. R. Campbell, F. M. Phillips, and B. D. Newman (2006), Isotopic exchange between snow and atmospheric water vapor: Estimation of the snowmelt component of groundwater recharge in the southwestern United States, J. Geophys. Res., 111, D09302, doi:10.1029/2005JD006470.

Angoran Baudelaire N'da, Lhoussaine Bouchaou, Barbara Reichert, Lahoucine Hanich, Yassine Ait Brahim, Abdelghani Chehbouni, El Hassane Beraaouz, Jean-Luc Michelot, 2016. Isotopic signatures for the assessment of snow water resources in the Moroccan high Atlas mountains: contribution to surface and groundwater recharge. Environmental Earth Sciences, 75:755, DOI: 10.1007/s12665-016-5566-9

Penna, D., van Meerveld, H.J., Zuecco, G., Dalla Fontana, G., Borga, M., 2016. Hydrological response of an Alpine catchment to rainfall and snowmelt events. Journal of Hydrology 537, 382–397. doi:10.1016/j.jhydrol.2016.03.040

Shanley, J. B., Kendall, C., Smith, T. E., Wolock, D. M. and McDonnell, J. J. (2002), Controls on old and new water contributions to stream flow at some nested catchments in Vermont, USA. Hydrol. Process., 16: 589–609. doi: 10.1002/hyp.312

---

## Referee Comment (RC2) · S. Pohl (Referee) · 30 Jun 2016

The authors present a study aimed at improving the reliability of isotopic hydrograph separation by including the spatial and temporal variability of snowmelt rates and isotopic composition in a high elevation basin. The authors use a very interesting dataset of isotopic measurements of a variety of components (snow, snow meltwater, stream water etc.) along with a state of the art hydrologic snowmelt model and some standard isotope hydrograph separation techniques to show the contribution of event and pre-event water to basin runoff during two snowmelt periods, one in early spring and one at the peak of snowmelt.

The topic is very interesting for a large research community as isotope studies of snow

processes have just begun to show their usefulness for a variety of scientific purposes but especially for the separation of hydrographs and for estimating the amount of snow meltwater to the overall river runoff. It therefore presents a really nice and important contribution to the scientific knowledge in the field of snow and isotope hydrology. The paper is written very well. The methods and techniques are appropriate and well applied. The study objectives are outlined clearly and the analysis for the most part follows these objectives. The results are presented in great detail in well readable Figures and Tables. The conclusions are based on the presented results and therefore well supported. The discussion is appropriate and commendably includes a clear segment stating the limitations of the study. The topic falls well within the scope of the journal and as stated will be a very important source of information for other researchers especially if they are attempting similar studies. There are only a few general and some specific comments that I recommend to address before the paper could be published.

General Comments:

One of the stated study objectives is "the estimation of the spatio-temporal variability of snowmelt and its isotopic content". I would argue that this part of the study could be enhanced and presented in more detail. The authors have acquired a quite unique dataset on the isotopic content at different times and at different places. The authors have acquired a quite unique dataset on the isotopic content at different times and at different places. Yet often the presented results are lumped together. This is the case for the spatial variability and for the temporal variability. For example, there are two north and two south facing sample points separated by roughly 400 m of elevation and 4000 m of horizontal distance. Yet unless I missed it all the isotope results are lumped together into "North" and "South". I think it would be very useful to present the results separately so that the reader can get a feeling for how much spatial variability there is within similar land surface classes but at different elevations or parts of the basin. This would help tremendously if one were to set up a similar study in another basin. The same can be said for the temporal variability. All the "sub-daily" samples seem to have

been lumped together into daily samples. Again a more detailed presentation of the data would be very interesting here.

The authors present and discuss the scenarios "North and South" in their IHS analysis. While I would agree that a short mention and presentation of the results of these two scenarios is helpful, I would keep this and any discussion of these scenarios very short, probably shorter than the authors have done. The reason is that no respectable researcher would or should attempt an IHS analysis using only samples from north or south facing slopes (certainly not after reading this study). The scenarios should therefore be considered purely theoretical and the authors should maybe focus the discussion more on the results obtained with the actually viable scenarios VWS, VWO, and VWE.

Specific Comments:

P.1 Line 21: I'm not sure I totally understand which methods are described here in respect to the north and south facing slopes.

P.1 Line 35: You might want to explain what water the term "pre event" refers to when it comes to studies of snowmelt contribution to runoff. Is this water stored in the soil or rock, i.e. is it purely groundwater or ground and soil water?

P.3 Line 34: Is the "Rofen valley area" identical to the Rofenach catchment? If so maybe use that term, otherwise restate the extent of glacial areas within the study catchment.

P. 4 line 8 If you want to refer to Figure 7 here you should reorder the sequence of Figures. I strongly believe that Figures should appear in the order in which they are addressed in the text of the paper.

P.4 Line 16 What are "sub daily grab samples" How many samples, temporal resolution and were the samples analyzed individually or combined as bulk samples?

P.4 Line 26 Were the snow pit layer samples used to eventually calculate weighted mean snow values using the layer thickness?

P. 5 Line 14 You might want to refer the reader to the section where the results of the model validation are actually shown.

P.5 Line 30 You subdivide the whole basin in either north or south with no class in between. While you state, that the valley runs mostly east-west and therefore the slopes and the DEM grids are mostly south or north, it would be good to show this visually, maybe by providing a graph showing the distribution of the grid aspects.

P.7 Line 5 "reflect" should probably be "reflecting"

P.8 first paragraph: You should briefly describe what are the main findings of Figure 7.

P.8 Line 14-16 The differences in the melt rates on north facing slopes during the early melt event are quite large. You might want to spend a little more time explaining these as the modelled values are quite important for the following analyses.

P.8 Line 36 should be "and could not clearly be obtained"

p.9 line 28 The authors state: "The hydrological response followed the diurnal variations of air temperature …. Because the available net-shortwave energy mostly controls the magnitude of snowmelt" This statement is not correct as it is. The diurnal air temperature variations have no control on the amount of net shortwave energy. It just so happens that the diurnal variations of air temp are usually fairly similar to those of net shortwave, but they do not influence each other. Please restate.

p. 10 line 22. See general comments: Was there no altitudinal gradient or did the authors just not discuss it?

p.10 line 30 You might want to replace "through" with "due to".

p.11 line 9 Maybe you should quickly list the assumptions (bullet points). They are all addressed in the following paragraph, but this would make it easier for the reader to understand what assumptions the authors are talking about.

p. 13 line 33. There are two definitions of the term "glacier melt". Sometimes snow

melting on a glacier is included in the term glacier melt, sometimes only ice melt is included. Please specify.

Figures 4 and 5: Maybe a boxplot graph would be a better idea to present the data.

Figure 8: There are fairly large differences in the observed vs. simulated snowmelt especially early on the north facing slope. In the text these differences are dealt with rather briefly. Maybe a slightly expanded discussion and explanation would be useful.

---

## Author Comment (AC1) · 4 Aug 2016

Dear referee #1,

thank you very much for your detailed review of the manuscript. Your comments have really helped to improve the quality of the paper. Please find our detailed answers to all referee comments (cumulative) in the attached .pdf-file (see supplements).

Best regards,

the authors

Please also note the supplement to this comment:

[Figure]

http://www.hydrol-earth-syst-sci-discuss.net/hess-2016-128/hess-2016-128-AC1-supplement.pdf

---

## Author Comment (AC2) · 4 Aug 2016

Authors reply to referee comments on "The importance of spatio-temporal snowmelt variability for isotopic hydrograph separation in a high-elevation catchment" by Schmieder et al. (hess-2016-128)

**Referee #1**

**Specific comments**

Comments to the Authors:

2, 32-35. This is a critical part of the Introduction and should be better explained. It seems to me that what stated here is more relevant for the temporal variability rather than the spatial variability. Please, specify.

Authors comment: Thank you very much for that hint. Yes, we totally agree and we will specify it in the manuscript.

**Comments to the Authors:**

3, 17-18. The third objective seems to me more a tool than a specific objective itself. I suggest to revise it.

**Authors comment:**

We will rewrite it in the revised manuscript.

**Comments to the Authors:**

4, 22-24. This is interesting, and I congratulate the authors for having collected both bulk snowmelt samples and sub-daily samples to assess the diurnal variability of snowmelt. However, as far as I see, no data are presented or no discussion is reported to compare the bulk with the sub-daily samples. I encourage the authors to do so because, in my opinion, knowing which variability we miss is we sample only once at the end of the day (bulk sample) instead of taking more samples during an individual melt event at the daily scale would be of great practical interest.

**Authors comment:**

Thank you, this is a good point. Unfortunately we couldn't sample sub-daily melt data at each location for each event. Because of this, and since the differences were small, we didn't expand the analyses so far. We will compare bulk values with single (sub-daily) values for the different sites where we have data and update the manuscript accordingly. We will use a separate figure to show the comparison graphically in detail. However, we want to emphasize, that it is not important for this specific study in our opinion to sample sub-daily, because we are interested in the event/daily scale, not the sub-daily scale.

**Comments to the authors:**

4, 6. According to Figure 2, Table 3 and 4, samples were collected (and hydrograph separation was conducted) for two snowmelt events in the early melt season (23 and 24 of April) and two snowmelt events in the late melt season (7 and 8 of June) but the authors talk about 'two short-term melt events (3 days)'. This is not clear because

usually diurnal-melt driven fluctuation in discharge are considered as individual melt events, so I count here 4 runoff events. More importantly, results are often presented in term of 'early melt' and 'peak melt', so, I believe, averaging or integrating the two couples of events. This operation could partly mask the intrinsic variability of the 4 events and therefore I suggest to present the data and the results separately (as, for instance, in Fig. 4).

**Authors comment:**

The focus of the study was on the inter-event variability and not that much on the intra-event variability. We think that it could be too little data for such an analysis. We agree that different definitions of events occur in the literature, but as mentioned above, we do not want to compare single days. So we will add a paragraph where we will describe our definition of the events in more detail, i.e. clear sky weather, warm and precipitation-free period with dry antecedent conditions for a few days during the entire snowmelt season (dry and warm spell).

**Comments to the Authors:**

4, 15-16. Two samples only to characterize baseflow and therefore the isotopic composition of pre-event water can be too little and a potential weak point for the following calculations. This aspect should be briefly discussed in the Discussion section or in section 5.6. Moreover, a recent work analyses the impact of sampling strategy of prevent water (before the individual melt event or before the start of the freshet period) for two-component hydrograph separation during snowmelt periods. The discussion about characterization of the pre-event water isotopic composition could start from the results obtained in the recent paper by Penna et al. (2016) (see suggested additional references below). Please note that only one sample of pre-event water is visible in Fig. 3: are the two samples so similar?

**Authors comment:**

Thank you for this suggestion. The two samples lie very close together (-15.05 ‰, -15.00 ‰), the difference is less than the lab precision, and we will specifically mention that in the manuscript. Therefore we argued that is likely not much variability in the baseflow isotopes. We checked this assumption with a dataset in the following winter baseflow season, where we investigated very small temporal variability between December and March. However, we suggest to not show these (because it would be post-event winter baseflow samples) results in this study, because they will be part of another study which is presently in preparation. We will add a small paragraph in the discussion, where we discuss the small sample number in comparison with the study from Penna et al. (2016). We plotted the average baseflow  $\overline{\delta}^{18}$ O value in Fig. 3, because this values was used for further analyses, but we will change it and plot the two single values individually.

**Comments to the authors:**

6, 23-24. The diurnal temporal variation of snowmelt isotopic signal (0.5 per mil) was used for the uncertainty of the event component in the hydrograph separation, according to the traditional approach by Genereux (1998). This is fine but one aspect is not clear to me: was the same variability used for each of the 4 events? I think it is very unlikely that the 4 of them have the same diurnal temporal variability. Instead, the variability of each day should be used for the assessment of uncertainty of each runoff event. Please, fix this or explain.

**Authors comment:**

We will change the manuscript and explain this in more detail to make clear that the uncertainty value is constituted of the diurnal variation (standard deviation) of snowmelt  $\delta^{18}$ O combined with the appropriate value of the two tailed t-table (dependent on sample number) as proposed by Genereux (1998). We used different uncertainty values for the early melt ( $W_{ce}$ =0.2 ‰) compared to the peak melt event ( $W_{ce}$ =0.5 ‰). This was not mentioned so far in the manuscript and we only reported the maximum uncertainty (i.e.  $W_{ce}$ =0.5 ‰). Unfortunately we do only have intra-daily  $\delta^{18}$ O snowmelt values for one site for one day per event because more sampling was not feasible. Thanks for pointing this out.

**Comments to the authors:**

6, 24. Here, the standard deviation of the two baseflow samples is relative small and it reflects, I assume, in small uncertainty values. I wonder if, having many baseflow samples, the variability would be greater and so the uncertainty of the pre-event component (see my comment above on 4, 15-16).

**Authors comment:**

That is a great idea and we will consider this idea in our future research. As mentioned before this is part of a future study which is already in preparation.

**Comments to the authors:**

11, 17-18. This is, to me, contrasting to what stated at 4, 22-24. I think that while the bulk sample integrate the diurnal melting cycle it also smooths out the variability of the snowmelt signal very much. This should be tested, reported and discussed.

**Authors comment:**

Thanks for this comment. In deed the sample rate smoothes out the sub-daily variability, but integrates it compared to a single snowmelt measurement at one time. That is what the respective paragraph in the manuscript was meant to describe. We used the bulk data for this, because we couldn't sample with the same sub-daily intervals/frequencies at each site. Because we were interest at the event scale, and the assumed maximum residence times of 24 hours, we used the bulk data for IHS analyses and used the daily variation for the uncertainty. We will add a paragraph in the discussion section of the revised version of the paper about that issue.

**Comments to the authors:**

Terminology: I recommend to consistently change the term 'isotopic content' into the more physically appropriate 'isotopic composition'. I also suggest to replace the term 'isotopic hydrograph separation' into 'isotope-based hydrograph separation'.

**Authors comment:**

We will change those terms according to the suggestions of the reviewer in the revised version of the manuscript.

**Comments to the authors:**

Figures 3. Use different colours or symbols for snowmelt and snowpack to better distinguish them. One pre-event samples is missing. Moreover, since there is an equation for the local meteoric water line, I suggest to plot it instead of the global meteoric water line...it makes more sense.

**Authors comment:**

Thank you, we will incorporate the suggestions of the reviewer in the revised version of the manuscript.

**Comments to the authors:**

Figure 4 and Figure 5. They are quite clear but I think that the information could be conveyed much more clearly by using box-plots instead. Please, consider changing these '1-D scatterplots' (by the way, is this the right term?) into box-plots.

**Authors comment:**

We used boxplots in the submitted version of the paper, but changed it to the 1dscatterplots as a result of a discussion with the editor. The idea was to show the distribution of the data more clearly which would be masked by the boxes. We added the median as a measure to compare the samples statistically. Furthermore some of the datasets are too small (n<5) for a boxplot with statistical validity. We used the term 1d scatterplot to emphasize the one dimensionality, so that it becomes clear the jitter arises randomly and has no meaning. It is also called jittered scatterplot or dotplot in the literature. However, we would be thankfull of a suggestion by the editor at this point for the right term.

**Comments to the authors:**

Figure 6 and Figure 7 are too small but this is probably due to the editorial form.

**Authors comment:**

We will provide the figures in a useful format and resolution in the revised version of the paper.

**Comments to the authors:**

Table 1. If Figure 4 and Figure 5 are converted to box-plots this table could be probably skipped because redundant. Please, consider this possibility.

**Authors comment:**

Thank you for bringing this up. As we agreed with the editor to use the dotplot figures, we will keep the table in the updated version too, in order to properly show the exact data values. If the editor suggests to revise the plot format we of course offer to change the manuscript according to the reviewers suggestion.

**Comments to the authors:**

Table 2. I think that it would be more informative to report the values of the two prevent samples individually. As I stated before, reporting the average of streamflow during the two early melt and the two late melt events is not so informative to me. Consider reporting all data in a different way (box-plots again) or even skipping this table and reporting the values of the two pre-event samples in the text.

**Authors comment:**

We will include the two pre-event values (-15.00 ‰, -15.05 ‰) in the text of the revised version of the paper. However, we still want to keep the table because of statistic overview of the data. We think that showing the average of the two events is informative because you can see that the values differ from each other. We thought it

could be useful for other scientists in the field to see the exact data values in table format.

Comments to the authors:

Table 3. Could this table be incorporated as bar plot in Figure 8? Please, consider the feasibility of this suggestion.

**Authors comment:**

We think that it could be generally a great idea to plot this information together and show all melt data (simulated and observed together), but in this case we think it would be hard to read the figure because different sites mix with distributed melt rates and there are different time scales making it hard to mix those dimensions. Furthermore, Table 3 shows SWE (not the melt rates) so we want to keep the table to show the benefit of having an understanding how much snow water equivalent is around.

Comments to the authors:

Table 5. Why is there no uncertainty reported for the peak event water fraction? According to Genereux (1998) it can be computed. Please, fix this.

Authors comment:

We will report the uncertainty for the peak water fraction according to the suggestion of the reviewer in the revised version of the manuscript. Thanks for pointing us in this direction.

**Minor comments and technical corrections**

Comments to the authors: 1, 11. I suggest to remove 'unknown'.

Authors comment: We will replace by with the term 'limited knowledge'.

Comments to the authors: 2, 39. Explain shortly which are the mentioned shortcomings.

Authors comment:

We will add a short explanation of the shortcomings VWA and CMW which are the exclusion of residence times.

Comments to the authors:

3, 7. For consistency, 'describe' should be 'described'.

Authors comment: Will be changed in the revised version of the manuscript.

Comments to the authors:

12, 15. Typo: 'were' should be 'where'.

**Authors comment: Will be changed in the revised version of the manuscript.**

Comments to the authors: 12, 29. The title is too long, please revise.

Authors comment:

The current title is the result of a long discussion among the authors. We think it is hardly possible to shorten the title without loosing the key message of the research presented in this publication. In this case a shorter title could carry the risk of being too general.

Comments to the authors: 13, 2. Is 'deployed' the right term here?

Authors comment: We will changed that term to "applied".

**Referee #2**

**General comments**

**Comments to the Authors:**

Yet often the presented results are lumped together. This is the case for the spatial variability and for the temporal variability. For example, there are two north and two south facing sample points separated by roughly 400 m of elevation and 4000 m of horizontal distance. Yet unless I missed it all the isotope results are lumped together into "North" and "South". I think it would be very useful to present the results separately so that the reader can get a feeling for how much spatial variability there is within similar land surface classes but at different elevations or parts of the basin. This would help tremendously if one were to set up a similar study in another basin. The same can be said for the temporal variability. All the "sub-daily" samples seem to have been lumped together into daily samples. Again a more detailed presentation of the data would be very interesting here.

**Authors comment:**

We thank the reviewer for this comment. There is unfortunately only one sample point at the south-facing and north-facing slope per event, respectively. The lower sample points were used (and sampled) for the early melt event, when snow cover was complete in the lower part of the catchment. The upper sample points were used for the peak melt event (when the snowline was higher). Spatial variability is the difference between north-facing and south-facing slopes, i.e. between the two sampling points. Elevation as spatial variability was not accounted for, but discussed in the discussion part. Sub-daily melt data was not presented because we do not have sub-daily melt data for each location and for each event/day. Therefore we used the bulk melt of one melt day for the analyses. Sub-daily snowmelt samples will be shown in a separate new figure for the locations where we have data (i.e. two sites for one day) to unravel those lumped information (compare comment from ref #1).

**Comments to the Authors:**

The authors present and discuss the scenarios "North and South" in their IHS analysis. While I would agree that a short mention and presentation of the results of these two scenarios is helpful, I would keep this and any discussion of these scenarios very short, probably shorter than the authors have done. The reason is that no respectable researcher would or should attempt an IHS analysis using only samples from north or south facing slopes (certainly not after reading this study). The scenarios should therefore be considered purely theoretical and the authors should maybe focus the discussion more on the results obtained with the actually viable scenarios VWS, VWO, and VWE.

**Authors comment:**

We totally agree with the statement of the reviewer. However, we want to emphasize on the importance of taking (more) samples at different slopes (with diverging aspects), and want to show the effects on IHS. Therefore we stress this hypothetical scenario, because it has not clearly been described in the literature before (as far as we know). We will address this issue more clearly in the revised version of the paper.

**Specific comments**

**Comments to the authors:**

P.1 Line 35: You might want to explain what water the term "pre event" refers to when it comes to studies of snowmelt contribution to runoff. Is this water stored in the soil or rock, i.e. is it purely groundwater or ground and soil water?

**Authors comment:**

We will add the term 'winter baseflow' in the description.

**Comments to the authors:**

P.3 Line 34: Is the "Rofen valley area" identical to the Rofenach catchment? If so maybe use that term, otherwise restate the extent of glacial areas within the study catchment.

Authors comment: Thank you for that hint. Yes, it is identical and we will exchange the term in the revised manuscript.

**Comments to the authors:**

P. 4 line 8 If you want to refer to Figure 7 here you should reorder the sequence of Figures. I strongly believe that Figures should appear in the order in which they are addressed in the text of the paper.

**Authors comment:**

Thank you for that comment, we simply overlooked it. We will change the order of the figure in the new version of the manuscript.

**Comments to the authors:**

P.4 Line 16 What are "sub daily grab samples" How many samples, temporal resolution and were the samples analyzed individually or combined as bulk samples?

Authors comment: We will expand the description at this section.

Comments to the authors:

P.4 Line 26 Were the snow pit layer samples used to eventually calculate weighted mean snow values using the layer thickness?

Authors comment: Yes, thank you. We will add this to that section.

Comments to the authors:

P. 5 Line 14 You might want to refer the reader to the section where the results of the model validation are actually shown.

Authors comment: We will refer to Section 4.2 in the revised manuscript.

Comments to the authors:

P.5 Line 30 You subdivide the whole basin in either north or south with no class in between. While you state, that the valley runs mostly east-west and therefore the slopes and the DEM grids are mostly south or north, it would be good to show this visually, maybe by providing a graph showing the distribution of the grid aspects.

Authors comment:

We have thought about this before but decided to not plot it, because the information content could be too little for a single figure, and it is already described in the text. We understand that it would be nice to see it visually (in a figure). If the editor thinks it is necessary to show it, we can of course do a subplot in Figure 1.

Comments to the authors:

P.7 Line 5 "reflect" should probably be "reflecting"

Authors comment:

This will be changed to 'reflecting'.

Comments to the authors:

P.8 first paragraph: You should briefly describe what are the main findings of Figure.

Authors comment:

We will add a briefly description of the main findings in Figure 7, i.e. mainly the underestimation of the simulated snow cover compared to the observed (MODIS/Landsat) snow cover.

Comments to the authors:

P.8 Line 14-16 The differences in the melt rates on north facing slopes during the early melt event are quite large. You might want to spend a little more time explaining these as the modelled values are quite important for the following analyses.

Authors comment: We will expand the explanation to make it more clearly. Comments to the authors: P.8 Line 36 should be "and could not clearly be obtained"

Authors comment: This will be changed in the revised manuscript.

**Comments to the authors:**

p.9 line 28 The authors state: "The hydrological response followed the diurnal variations of air temperature . . .. Because the available net-shortwave energy mostly controls the magnitude of snowmelt" This statement is not correct as it is. The diurnal air temperature variations have no control on the amount of net shortwave energy. It just so happens that the diurnal variations of air temp are usually fairly similar to those of net shortwave, but they do not influence each other. Please restate.

Authors comment: We will restate the sentence according to the suggestions of the reviewer.

Comments to the authors:

p. 10 line 22. See general comments: Was there no altitudinal gradient or did the authors just not discuss it?

Authors comment:

Unfortunately we do not have data to reveal an altitude gradient, but we discussed a hypothetical scenario, i.e. how would a decrease of snowmelt  $\delta^{18}$ O with altitude affect IHS results.

Comments to the authors:

p.10 line 30 You might want to replace "through" with "due to".

Authors comment:

This will be changed in the revised manuscript.

Comments to the authors:

p.11 line 9 Maybe you should quickly list the assumptions (bullet points). They are all addressed in the following paragraph, but this would make it easier for the reader to understand what assumptions the authors are talking about.

Authors comment:

Thank you for this suggestion. We also think that it will be a step towards an improved readability of the manuscript. Therefore we will incorporate the listed assumptions in the manuscript.

Comments to the authors:

p. 13 line 33. There are two definitions of the term "glacier melt". Sometimes snow melting on a glacier is included in the term glacier melt, sometimes only ice melt is included. Please specify.

Authors comment: We will specify it in the revised manuscript. Comments to the authors:

Figures 4 and 5: Maybe a boxplot graph would be a better idea to present the data.

Authors comment: see authors response to referee #1

We used boxplots in the submitted version of the paper, but changed it to the 1dscatterplots as a result of a discussion with the editor. The idea was to show the distribution of the data more clearly which would be masked by the boxes. We added the median as a measure to compare the samples statistically. Furthermore some of the datasets are too small (n<5) for a boxplot with statistical validity. We used the term 1d scatterplot to emphasize the one dimensionality, so that it becomes clear the jitter arises randomly and has no meaning. It is also called jittered scatterplot or dotplot in the literature. However, we would be thankful of a suggestion by the editor at this point for the right term.

Comments to the authors:

Figure 8: There are fairly large differences in the observed vs. simulated snowmelt especially early on the north facing slope. In the text these differences are dealt with rather briefly. Maybe a slightly expanded discussion and explanation would be useful.

Authors comment: We will expand on this in the appropriate section of the manuscript.

---

## Author Response (AR2)

[revised manuscript text omitted]

Authors comment:
We thank the reviewer as well as the editor for their helpful suggestions and think
that the overall quality of the manuscript as well as the readability has improved a lot.
We changed the suggested comments in the revised version of the manuscript
including minor comments, title change, wordings and the updated reference list. The
changes are highlighted in yellow in the marked up manuscript. Thank you very much
for all your endeavors.

---

## Editor Decision (ED2)

**1 The importance of snowmelt spatio-temporal variability for**

**2 isotope-based hydrograph separation in a high-elevation 3 catchment**

Jan Schmieder1, Florian Hanzer1, Thomas Marke1, Jakob Garvelmann2, Michael Warscher2,
 Harald Kunstmann2 and Ulrich Strasser1

6 1Institute of Geography, University of Innsbruck, Innsbruck, 6020, Austria

[revised manuscript text omitted]

- 26

---

## Author Response (AR3)

**Authors reply to referee comments on "The importance of snowmelt spatio-temporal variability for isotope-based hydrograph separation in a high-elevation catchment" by Schmieder et al. (hess-2016-128)**

Authors comment:
We implemented all suggested technical corrections and would like to thank the editor for her helpful suggestions and think that the readability as well as the wordings has improved further. The changes are highlighted in yellow in the marked up manuscript. Thank you very much for all your endeavors.

[revised manuscript text omitted]